# Multi-Modal Knowledge Distillation for Recommendation with Prompt-Tuning

## ABSTRACT

Multimedia online platforms (*e.g.*, Amazon, TikTok) have greatly benefited from the incorporation of multimedia content (*e.g.*, visual, textual, and acoustic modalities) into their personal recommender systems. These modalities provide intuitive semantics that facilitate modality-aware user preference modeling. However, two key challenges in multi-modal recommenders remain unresolved: i) The introduction of multi-modal encoders with a large number of additional parameters causes overfitting, given high-dimensional multi-modal features provided by extractors (*e.g.*, ViT, BERT). ii) As side information, media content inevitably introduces inaccuracies and redundancies, which skew the modality-interaction dependency from reflecting true user preference. To tackle these problems, we propose to simplify and empower recommenders through Multimedia Knowledge Distillation (MMKD) with the prompt-tuning that enables adaptive quality distillation. Specifically, MMKD conducts model compression through distilling u-i edge relationship and multi-modal node content from cumbersome teachers to relieve students from the additional feature reduction parameters. To bridge the semantic gap between multi-modal context and collaborative signals for empowering the overfitting teacher, soft prompt-tuning is introduced to perform student task-adaptive distillation for tackling redundancy. Additionally, to adjust the impact of inaccuracies in multimedia data, a disentangled multi-modal list-wise distillation is developed with modality-aware re-weighting mechanism. Experiments on real-world data demonstrate MMKD's superiority over existing techniques. Ablation tests confirm the effectiveness of key components. Additional tests show the efficiency and effectiveness.

**ACM Reference Format:**
Anonymous Author(s). 2023. Multi-Modal Knowledge Distillation for Recommendation with Prompt-Tuning. In *Proceedings of ACM Conference (Conference'17)*. ACM, New York, NY, USA, 12 pages. https://doi.org/10.1145/nnnnnnn.nnnnnnn

## 1 INTRODUCTION

Multimedia platforms (*e.g.*, Tiktok, Amazon) have grown in importance as tools for sharing and shopping online. Utilizing modalities (*e.g.*, soundtracks of videos, pictures of products) to identify customized user preferences for item ranking is the target of multi-modal recommendation system [5, 46]. Early works started with introducing visual content[9, 16] into Collaborative Filtering(CF)[18] framework and later works employed attention mechanism for tailored signals[35, 70]. After that, graph neural network (GNNs) methods (*e.g.*, MMGCN[64], GRCN [63]) became mainstream due to significant improvement by modeling high-order[58] relations. Based on previous progress, some multi-modal works focus on alleviating sparsity issue by constructing homogeneous graphs(*e.g.*,

u-u[55], i-i[61, 74]) or introducing self-supervised tasks through joint training(*e.g.*, CLCRec [62], SLMRec [53], and MICRO [75]).

Despite the progress made in previous works, some key **issues** still remain explored for multimedia recommendation scenarios:

- **I1: Overfitting & Sparsity.** Current multimedia recommenders excel by employing advanced encoders to handle high-dimensional features from pre-trained extractors (CLIP-ViT[47], BERT[3]). The auxiliary modalities were originally intended to alleviate data sparsity, but actually inevitably lead to increased consumption [56]. For example, regarding feature extractors of Electronics (Sec. 4.1.1) dataset, the output dimension of SBERT[48] and CNNs[22] are 768 and 4,096, respectively. They are much larger than embedding dimensions of current methods[62, 64], *i.e.*, $d_m \gg d$. Retraining pre-trained models can change output dimensions, but will significantly impact performance due to different latent representations and hyperparameters. Besides, training pre-trained models demands significant computational resources and can take days to weeks on multiple GPUs. Therefore, current multi-modal works[5, 60] carry additional high-dimensional feature reduction layers. These additional parameters aggravated overfitting that already exists due to data sparsity, further increasing the difficulty of convergence and leading to suboptimal results[59].

- **I2: Noise & Semantic Gap.** As side information, multimedia content has inherent inaccuracies and redundancies when assisting user preference learning with collaborative relations. For example, a user may be attracted by a textual title, but the image content is unrelated; and the music in micro-videos might be for trends, not user preferences. Blindly relying on noisy modality data may mislead the u-i relation modeling. Besides, the multi-modal context and u-i collaborative relations are originally derived from two different distributions with a large semantic gap [60], which poses challenges in mining modality-aware user preference and even disrupts the existing sparse supervisory signals.

To cope with the above issues, we propose the following solutions: **I1:** Developing a multi-modal knowledge distillation (MMKD) recommendation framework to free the inference recommender from the additional feature reduction parameters, by using KD for model compression. This paradigm prevents overfitting while maintaining accuracy, which also boosts the critical online inference phase with fewer resources. Specifically, MMKD conducts model compression through distilling edge relationship (ranking KD, denoised modality-aware ranking KD), and node content (modality-aware embedding KD). The three types of KD respectively convey i) Pure knowledge through a modified KL divergence[29] based on BPR loss[49]; ii) Fine-grained modality-aware list-wise ranking knowledge; iii) Modality-aware embedding KD through SCE loss [20], an enhanced version of MSE. **I2:** Developing two modules to tackle issues '*Noise & Semantic Gap*' based on the KD framework: i) Semantic bridging soft prompt-tuning is meant to reduce the impact of redundancy by prompting teacher to deliver student-task adaptive knowledge. In

*Conference'17, July 2017, Washington, DC, USA*
2023. ACM ISBN 978-x-xxxx-xxxx-x/YY/MM. . . $15.00
https://doi.org/10.1145/nnnnnnn.nnnnnnn

Table 1: Model compression analysis. Time complexity comparison among SOTA GNN-enhanced multi-modal recommenders. i) the $\mathcal{R}(\cdot)$: time complexity of multi-modal feature reduction layer, by mapping high-dimensional features into dense embeddings, i.e., $d_m \rightarrow d$. ii) the $GNNs$: time complexity of various GNN architectures in different models for message propagation.

| Component | MMGCN [64] | GRCN [63] | LATTICE [74] | SLMRec [53] | BM3 [80] | MMKD |
|---|---|---|---|---|---|---|
| $\mathcal{R}(\cdot)$ | $O(\sum\limits_{m\in M}\|\mathcal{I}\|(d_m+d)d_h)$ | $O(\sum\limits_{m\in M}\|\mathcal{I}\|d_m d)$ | $O(\sum\limits_{m\in M}\|\mathcal{I}\|d_m d)$ | $O(\sum\limits_{m\in M}\|\mathcal{I}\|d_m d)$ | $O(\sum\limits_{m\in M}\|\mathcal{I}\|d_m d)$ | 0 |
| GNNs | $O(\sum\limits_{m\in M}L\|\mathcal{E}\|d^3)$ | $O(\sum\limits_{m\in M}(\|\mathcal{I}\|^2 d + L\|\mathcal{E}\|d))$ | $O(\sum\limits_{m\in M}\|\mathcal{I}\|^2 d_m + k\|\mathcal{I}\|log(\|\mathcal{I}\|)+L\|\mathcal{E}\|d)$ | $O(\sum\limits_{m\in M}L\|\mathcal{E}\|d)$ | $O(L\|\mathcal{E}\|d)$ | $O(L\|\mathcal{E}\|d)$ |

other words, prompt-tuning module can bridge the semantic gap in two aspects: multi-modal content & collaborative signals, and student & frozen teacher. Technically, the module is incorporated into the teacher's reduction layer and constructs prompts based on multi-modal features. For optimization, the soft prompts train with both teacher and student, to adaptively guide students during the distillation when teacher is frozen. ii) Modality-aware disentangled denoising list-wise ranking KD is to adjust the influence of inaccuracies in modality-aware user preference. The decoupled KD process first separates the results of list-wise ranking based on modality-specific presentation. A re-weighting mechanism is then applied to adjust the influence of unreliable portions.

To summarize, the main contributions of this work are as follows:

- In this work, we propose a novel multi-modal KD framework MMKD for multimedia recommendation, which is able to produce a lightweight yet effective student inference recommender with minimal online inference time and resource consumption.

- We integrate prompt-tuning with multi-modal KD to bridge the semantic gap between modality content and collaborative signals. Additionally, by disentangling the modality-aware ranking logits, the impact of noise in multimedia data is adjusted.

- We conduct experiments to evaluate our model performance on real-world datasets. The results demonstrate our MMKD outperforms state-of-the-art baselines. The ablation studies and further analysis show the effectiveness of sub-modules.

To facilitate the result reproducibility, our implementation is released via link: https://anonymous.4open.science/r/MMKD.

## 2 PRELIMINARIES

**Interaction Graph with Multi-Modal Context**. Motivated by the effectiveness of graph-based recommenders, we represent user-item relationships as a bipartite graph $\mathcal{G} = (\{\mathcal{U}, \mathcal{I}\}, \mathcal{E}, \mathbf{X})$. Here, $\mathcal{U}, \mathcal{I}$ are users' set and items' set, respectively. The edges $\mathcal{E}$ in $\mathcal{G}$ can be represented by adjacency matrix $\mathbf{A} \in \mathbb{R}^{|\mathcal{U}| \times |\mathcal{I}|}$ with $\mathbf{A}_{[u,i]} = 1$ if the implicit feedback exists, otherwise $\mathbf{A}_{[u,i]} = 0$. Furthermore, each item $i \in \mathcal{I}$ is associated with multi-modal features $\mathbf{X}_i = \{\mathbf{x}_i^1, ..., \mathbf{x}_i^m, ..., \mathbf{x}_i^{|\mathcal{M}|}\}$, where $|\mathcal{M}|$ is the number of modalities, indexed by $m \in \mathcal{M}$. The feature $\mathbf{x}_i^m$ is a high-dimensional vector in $\mathbb{R}^{d_m}$ that captures the characteristics of modality $m$. Notably, the dimensions $d_m$ of multimodal features are often much larger than those $d$ of recommender representations, i.e., $d_m \gg d$.

**Task Formulation.** The goal of multi-modal recommender systems is to learn a function that predicts the likelihood of a user adopting an item, given an interaction graph $\mathcal{G}$ with multi-modal context $\mathbf{X}$. The output of the predictive function is the learned preference score of a target user $u$ over a non-interacted item $i$.

## 3 METHODOLOGY

MMKD conducts model compression to build a lightweight yet effective multi-modal recommender for resource-friendly online collaborative filtering. The overall model flow is shown in Fig. 1. Key components will be elaborated in following subsections.

### 3.1 Modality-aware Task-adaptive Modeling

*3.1.1* **Teacher-Student in CF.** Knowledge distillation aims to compress a complex large model into a lightweight and effective small model. Inspired by this, our developed MMKD is to transfer modality-aware collaborative signals from cumbersome teacher to lightweight student. For optimization, we employ offline distillation [11] which is a two-stage process, for flexibility concerns. In the first stage, only the teacher is trained, and in the second stage, the teacher remains fixed while only the student is trained.

**Teacher** $\mathcal{T}$ follows pattern of current graph-based multi-modal encoders [60, 74], which encodes id-corresponding embeddings $\mathbf{E}_u^{\mathcal{T}}, \mathbf{E}_i^{\mathcal{T}}$ and modality-specific features $\mathbf{F}_u^m, \mathbf{F}_i^m$ through GNNs. The two types of encoded representations will be further distilled to student by our modality KD in Sec. 3.2.2, Sec. 3.2.3 and collaborative KD in Sec. 3.2.1. Teacher $\mathcal{T}$ encoding process can be as follows:

$$\{\mathbf{E}_u^{\mathcal{T}}, \mathbf{E}_i^{\mathcal{T}}\}, \quad \{\mathbf{F}_u^1, ..., \mathbf{F}_u^m, ..., \mathbf{F}_i^1, ..., \mathbf{F}_i^m ...\} = \mathcal{T}(\mathbf{A}, \mathbf{X}) \quad (1)$$

The two types of outputs respectively convey reliable collaborative signals and modality-aware user preferences to student. $\mathbf{F}_u^m \in \mathbb{R}^{|\mathcal{U}| \times d}, \mathbf{F}_i^m \in \mathbb{R}^{|\mathcal{I}| \times d}$ are compressed (i.e., $d_m \rightarrow d$) from high-dimensional $\mathbf{X} \in \mathbb{R}^{|\mathcal{I}| \times d_m}$ from extractors (e.g., BERT [27]).

**Student** $\mathcal{S}$ utilizes lightweight LightGCN [17] to capture user-item collaborative relationship. The embedding process is conducted without computationally intensive encoding of multi-modal features. The encoding of student $\mathcal{S}$ can be summarized as:

$$\mathbf{E}_u^{\mathcal{S}}, \ \mathbf{E}_i^{\mathcal{S}} = \mathcal{S}(\mathbf{A}) \quad (2)$$

$\mathbf{E}_u^{\mathcal{S}}, \mathbf{E}_i^{\mathcal{S}}$ are the final user and item presentation used for online recommendation inference and for receiving teacher knowledge.

*3.1.2* **Soft Prompt-Tuning as Semantic Bridge.** Modality content $\mathbf{X}$ inevitably includes ranking task-irrelevant redundancies, which not only confuse the target CF task but also exacerbate overfitting. Besides, the large semantic gap between general-purpose modality modeling and u-i interaction modeling also hinders true user preferences. Drawing inspiration from parameter efficient fine-tuning (PEFT) [31, 33], we employ soft prompt-tuning[31] as the solution. Specifically, we incorporate prompt $\mathbf{p}$ into teacher $\mathcal{T}(\cdot)$'s multi-modal feature reduction layer $\mathcal{R}(\cdot)$, to facilitate the extraction of collaborative signals from modalities. $\mathbf{p}$ is constructed by multi-modal features $\mathbf{X}$ and finetuned with student $\mathcal{S}(\cdot)$ to provide the frozen teacher $\mathcal{T}(\cdot)$ a sutdent-task related signals as a hint. The specific process can be divided into three steps: i) Construct the prompt; ii) Incorporate in teacher $\mathcal{T}(\cdot)$; iii) Conduct prompt-tuning.

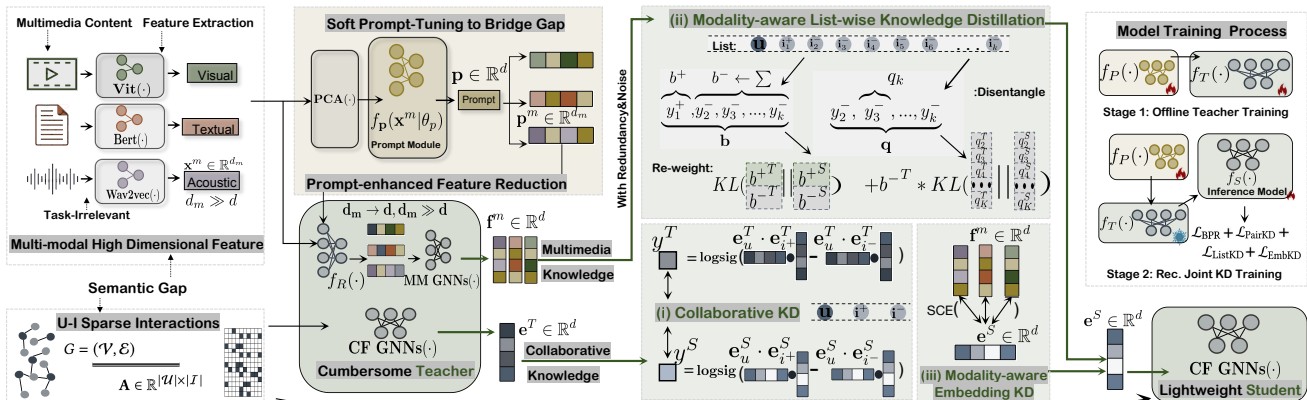

**Figure 1: MMKD is to learn a lightweight recommender with minimal online consumption, including three types of KD: i) ranking KD; ii) denoised modality-aware ranking KD; iii) modality-aware embedding KD. Besides, prompt-tuning is for adaptive task-relevant KD; disentangling and re-weighting are introduced to adjust the impact of noise in modalities.**

**Prompt Construction**. To better incorporate semantics to prompt module $\mathcal{P}(\cdot)$, we initialize $\mathbf{p}$ using semantic content[51], instead of vanilla initialization (*e.g.*, Xavier [10], uniform). Refer to Prefix-tuning[33], $\mathcal{P}(\cdot)$ is a feedforward layer takes soft prompt $\mathbf{p}$ as input which aggregates information from multi-modal item features $\mathbf{x}^m$. The process of obtaining prompt vectors is as follows:

$$\mathbf{p} = \mathcal{P}(\mathbf{x}^m|\theta_{\mathcal{P}}) = \mathcal{P}\left(\frac{1}{|\mathcal{M}|}\sum_{m\in\mathcal{M}}^{|\mathcal{M}|}\eta(\mathbf{x}^m)\right) \quad (3)$$

$\eta(\cdot)$ denotes the dimensionality reduction function (*e.g.*, PCA) for multi-modal features. The learned prompt $\mathbf{p}$ will be incorporated into the teacher's inference process. The soft prompt module $\mathcal{P}(\cdot)$ will offer adaptive cues to the teacher once the student $\mathcal{S}(\cdot)$ is trained and the teacher $\mathcal{T}(\cdot)$ is frozen.

**Prompt-guided Teacher.** Having obtained prompt $\mathbf{p}$, we apply it to the feature reduction layer $\mathcal{R}(\cdot)$ in teacher $\mathcal{T}(\cdot)$ for enhancing the overfitting teacher, while simultaneously conducting student-task adaptive knowledge distillation through the frozen teacher. To be specific, we transform our prompt $\mathbf{p}$ into modality-specific module, *i.e.*, $\mathbf{p} \to \mathbf{p}^m$, which allows the prompt to capture modality-specific information. Next, our method leverages a simple yet effective add operator, inspired by [21], to integrate the modality-specific prompt $\mathbf{p}^m$ into the teacher's multi-modal feature encoding layer. Formally, this prompt integration process can be given as follows:

$$\mathbf{f}^m = \mathcal{R}(\mathbf{x}^m, \mathbf{p}^m|\theta_{\mathcal{R}}) = \varrho((\mathbf{x}^m + \lambda_1 * \mathbf{p}^m)\mathbf{W}_{\mathcal{R}}^m + \mathbf{b}_{\mathcal{R}}^m) \quad (4)$$

$\mathcal{R}(\cdot)$ takes high-dimensional multi-modal features $\mathbf{x}^m$ and modality-specific prompts $\mathbf{p}^m$ as inputs, and output modality-specific embeddings $\mathbf{f}^m \in \mathbb{R}^d$. The modality-specific prompt $\mathbf{p}^m \in \mathbb{R}^{d_m}$ is obtained by reshaping (*i.e.*, $d \to d_m$) from $\mathbf{p}$ through $\mathbf{p}^m = \mathbf{p} \cdot \mathbf{p}^T \mathbf{x}^m$, and adjusted by factor $\lambda_1$. To prevent overfitting caused by numerous parameters high-dimensional features $\mathbf{x}^m$'s reduction, dropout $\varrho(\cdot)$ is applied here. The filter parameters $\mathbf{W}_{\mathcal{R}}^m$ and $\mathbf{b}_{\mathcal{R}}^m$ are used to map modality-specific features to their respective embedding space.

By this way, the feature reduction $\mathcal{R}$ will be strengthened due to: i) bridging the gap between modality content and collaborative signals, extracting modality-aware user preferences; ii) facilitating

**Table 2: Summary of Key Notations.**

| Notations | Explanations |
|---|---|
| $\mathcal{G}, \mathcal{V}, \mathcal{E}$ | Interaction graph, Node set, Edge set |
| $\mathbf{x}^m \in \mathbb{R}^{d_m}, \mathbf{f}^m \in \mathbb{R}^d$ | High dimensional/ Densified feature of $\mathcal{T}$ |
| $\mathbf{E}_u^{\mathcal{T}} \in \mathbb{R}^{\mathcal{U}\times d}, \mathbf{E}_u^S \in \mathbb{R}^{\mathcal{U}\times d}$ | Final user embedding of teacher/student |
| $\mathcal{T}(\cdot), \mathcal{S}(\cdot), \mathcal{P}(\cdot), \mathcal{R}(\cdot)$ | Teacher, Student, Prompt Module, Reduction |
| $\mathbf{b}, \mathbf{q}$ | Binarized/Re-weighted knowledge |
| $b^+/b^-, q_k$ | Binarized/Re-weighted single score |

* We use uppercase bold letters (*e.g.*, $\mathbf{X}$) to denote matrices, lowercase bold letters (*e.g.*, $\mathbf{x}$) to denote vectors and light letters to denote scalar values. Algorithm pseudo code in Supplementary.

knowledge distillation process by making modality-aware student constrained prompt $\mathbf{p}$ participate in teacher's inference.

**Soft Prompt-tuning Paradigm**. In KD's soft prompt tuning, we consider the cumbersome teacher as the pre-trained model, and the process is split into two stages. During teacher training, the prompt module $\mathcal{P}(\cdot)$ undergoes gradient descent with teacher $\mathcal{T}(\cdot)$, affecting the teacher's inference process. During student training, we employ offline knowledge distillation[11], freezing the teacher's parameters $\theta_{\mathcal{T}}$ and updating the prompt module $\mathcal{P}(\cdot)$ again according to the student's recommended loss, which allows the prompt $\mathbf{p}$ to provide additional guidance to the feature reduction process and distill task-relevant knowledge from teacher $\mathcal{T}(\cdot)$.

### 3.2 Modality & Ranking Knowledge Distillation

To comprehensively obtain quality collaborative signal and modality-aware user preference from teacher $\mathcal{T}(\cdot)$, we have designed three types of KD paradigms to convey knowledge from different perspectives: i) Ranking KD; ii) Denoised Modality-aware Ranking KD; and iii) Modality-aware Embedding KD.

*3.2.1* **Pure Ranking KD.** As a ranking task, teacher $\mathcal{T}(\cdot)$ ought to convey task-relevant collaborative relations. To this end, we propose to utilize prediction logits in ranking objectives such as BPR[49] for KD optimization. Specifically, we distill valid ranking knowledge from the ultimate representation $\mathbf{E}_u^{\mathcal{T}}, \mathbf{T}_i^{\mathcal{T}}$ constrained by the classical pair-wise ranking BPR loss. Pair-wise score $y_{\text{pair}}^{\mathcal{T}}$ and $y_{\text{pair}}^S$ are is taken as a logit of KD loss for teacher and student, respectively. The classic KL loss logit represents multi-class scores, while $y_{\text{pair}}$ represents a binary classification logit for determining

whether $i_+$ is better than $i_-$ for user $u$. Our KD paradigm with the pairwise ranking loss can be formally presented as follows:

$$\mathcal{L}_{\text{PairKD}}(\Theta_{\mathcal{S}};\Theta_{\mathcal{P}}) = -\sum_{(u,i^+,i^-)}^{|\mathcal{E}_{\text{bpr}}|} y_{\text{pair}}^{\mathcal{T}}(\log y_{\text{pair}}^{\mathcal{T}} - \log y_{\text{pair}}^{\mathcal{S}}) \quad (5)$$

$$y_{\text{pair}} = \log(\text{sigmoid}(\mathbf{e}_u \cdot \mathbf{e}_{i^+} - \mathbf{e}_u \cdot \mathbf{e}_{i^-}))$$

where $\mathcal{L}_{\text{PairKD}}$ represents the pair-wise ranking KD objective. $\theta_S; \theta_P$ means that both student $\mathcal{S}(\cdot)$ and prompt module $\mathcal{P}(\cdot)$ parameters are updated with the loss $\mathcal{L}_{\text{PairKD}}(\cdot)$. In each step, MMKD samples a batch of triplets $\mathcal{E}_{\text{bpr}} = \{(u, i^+, i^-)|\mathbf{A}_{[u,i^+]} = 1, \mathbf{A}_{[u,i^-]} = 0\}$, where $u$ denotes the target user. Here, $i^+$ and $i^-$ denotes the positive item and negative item of BPR loss, respectively. In this way, teacher model $\mathcal{T}(\cdot)$ imparts collaborative expertise to student model $\mathcal{S}(\cdot)$, offering rich implicit knowledge in a different solution space [28] to help the student escape from local optima [7, 14].

*3.2.2* **Denoised Modality-aware Ranking Disentangled KD.** Previously encoded multi-modal content $\mathbf{f}_u^m, \mathbf{f}_i^m$ in teacher $\mathcal{T}(\cdot)$ contains noise and can affect the modality-aware user preferences modeling. To conduct accuracy and fine-grained distillation while reducing the impact of task-irrelevant parts, we design a denoised modality-aware KD. Specifically, we calculate the list-wise score using $\mathbf{f}_u^m, \mathbf{f}_i^m$ to perform modality-aware ranking KD. In addition, to further reduce the impact of noise, we reformulate KD loss into a weighted sum of the disentangled parts.

**Disentangling Modality-aware List-wise Score.** For a $K$ samples ranking list, the predicted logits can be denoted as $\mathbf{y}_{\text{list}} = [y_1^+; y_2^-, y_3^-, ..., y_k^-, ..., y_K^-]$, where $y^+$ and $y^-$ are the scores of the observed edge $\mathbf{A}^+$ and unobserved edge $\mathbf{A}^-$, respectively. MMKD take each score in $\mathbf{y}_{\text{list}}$ as logit in KL divergence for distilling informative tacit knowledge[11]. The modality-aware list-wise logits then can be reformulated into two parts $KL(\mathbf{b}^{\mathcal{T}}\|\mathbf{b}^{\mathcal{S}})$ and $KL(\mathbf{q}^{\mathcal{T}}\|\mathbf{q}^{\mathcal{S}})$. $\mathbf{b}^{\mathcal{T}}$ deliver overall user preference to $\mathbf{b}^{\mathcal{S}}$; $\mathbf{q}^{\mathcal{T}}$ deliver fine-grained list-wise ranking prefer to $\mathbf{q}^{\mathcal{S}}$. More specifically, $\mathbf{b} = [b^+, b^-] \in \mathbb{R}^{1\times2}$ represents the binary logits of observed set $\{y_1^+\}$ and unobserved set $\{y_2^-, y_3^-, ..., y_k^-, ..., y_K^-\}$, that softened by softmax:

$$\mathbf{b}: \quad b^+ = \frac{exp(y_1^+)}{\sum_{k=1}^K exp(y_k)}; \quad b^- = \frac{\sum_{k=2}^K exp(y_k^-)}{\sum_{k=1}^K exp(y_k)} \quad (6)$$

Note that, $\hat{y}$ is the sum of unobserved sets. Meanwhile, we declare $\mathbf{p} = [\hat{y}_2, \hat{y}_3, ..., \hat{y}_k, ..., \hat{y}_K] \in \mathbb{R}^{1\times K}$ to independently model logits among unobserved set (i.e., without considering $y^+$). Each element is calculated by:

$$\mathbf{q}: \quad q_k = \frac{exp(y_k^-)}{\sum_{k=2}^K exp(y_k)} \quad (7)$$

**Re-weighting Modality-aware List-wise Score.** Afterward, lower scores are assigned to those uncertain user-item relationships to down-weight their influence in the KD process. This allows MMKD to focus on the most reliable signals from the teacher model for denoised knowledge transfer. The vanilla KL-Divergence can be disentangled and re-weighted through the following derivation [1]:

$$KL(\mathbf{y}_{\text{list}}^{\mathcal{T}}\|\mathbf{y}_{\text{list}}^{\mathcal{S}}) = b^{+\mathcal{T}}\log(\frac{b^{+\mathcal{T}}}{b^{+\mathcal{S}}}) + \sum_{k=2,i\neq1}^K y_k^{-\mathcal{T}}\log(\frac{y_k^{-\mathcal{T}}}{y_k^{-\mathcal{S}}}) \quad (8)$$

---
[1]We omit the temperature $\tau$ of softmax [19] without loss of generality

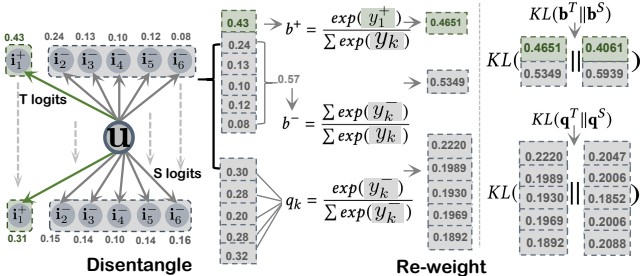

**Figure 2: Calculation Example of Disentangled KD**

According to Eq. 6 and Eq. 7, we can derive $q_k = y_k/b^-$. Thus, Eq. 8 can be rewritten as follows (detailed derivations in Appendix ):

$$= b^{+\mathcal{T}}\log(\frac{b^{+\mathcal{T}}}{b^{+\mathcal{S}}}) + b^{-\mathcal{T}}\sum_{k=2,i\neq1}^K q_k^{\mathcal{T}}(\log(\frac{q_k^{\mathcal{T}}}{q_k^{\mathcal{S}}}) + \log(\frac{b^{-\mathcal{T}}}{b^{-\mathcal{S}}})) \quad (9)$$

$$= \underbrace{b^{+\mathcal{T}}\log(\frac{b^{+\mathcal{T}}}{b^{+\mathcal{S}}}) + b^{-\mathcal{T}}\log(\frac{b^{-\mathcal{T}}}{b^{-\mathcal{S}}})}_{KL(\mathbf{b}^{\mathcal{T}}\|\mathbf{b}^{\mathcal{S}})} + (1 - b^{+\mathcal{T}})\underbrace{\sum_{k=2,i\neq1}^K q_k^{\mathcal{T}}\log(\frac{q_k^{\mathcal{T}}}{q_k^{\mathcal{S}}})}_{KL(\mathbf{q}^{\mathcal{T}}\|\mathbf{q}^{\mathcal{S}})}$$

Then, we can reformulate our disentangled knowledge distillation paradigm with the awareness of multi-modalities as follows:

$$\mathcal{L}_{\text{ListKD}} = -\sum_{m\in\mathcal{M}}^{|\mathcal{M}|} KL(\mathbf{b}^{\mathcal{T}}\|\mathbf{b}^{\mathcal{S}}) + (b^{+\mathcal{T}} - 1)KL(\mathbf{q}^{\mathcal{T}}\|\mathbf{q}^{\mathcal{S}}) \quad (10)$$

List-wise ranking KD loss $\mathcal{L}_{\text{ListKD}}$ is reformulated as a weighted sum of two terms for adjustablely transferring reliable knowledge and enhancing the accuracy of modality-relevant user preference.

*3.2.3* **Modality-aware Embedding Distillation.** In addition to the logit-based knowledge distillation, we propose to enhance our MMKD framework with embedding-level distillation. To achieve embedding alignment in our MMKD, we employ the Scale Cosine Error (SCE) [20] loss function for robust training instead of Mean Square Error (MSE). This is because MSE is sensitive and unstable, which can lead to training collapse [20] because of varied feature vector norms and the curse of dimensionality [6]. The utilization of the SCE-based loss $\mathcal{L}_{\text{EmbKD}}$ for embedding-level knowledge distillation can take the following forms:

$$\mathcal{L}_{\text{EmbKD}} = \sum_{m\in\mathcal{M}}^{|\mathcal{M}|}\frac{1}{|\mathcal{I}|}\sum_{i\in\mathcal{I}}(1 - \frac{\mathbf{e}_i^{\mathcal{S}} \cdot \mathbf{f}_i^m}{\|\mathbf{e}_i^{\mathcal{S}}\| \times \|\mathbf{f}_i^m\|})^{\gamma}, \gamma \geq 1 \quad (11)$$

$\mathcal{L}_{\text{EmbKD}}$ is averaged over all user and item nodes in the interaction graph $\mathcal{G}$. The final representation outputted by the student model $\mathcal{S}(\cdot)$ is denoted as $\mathbf{e}_i^{\mathcal{S}} \in \mathbf{E}_i^{\mathcal{S}}$, while the encoded multi-modal feature from the teacher function $\mathcal{T}(\cdot)$ are denoted as $\mathbf{f}_i^m \in \mathbf{F}_i^m$. The scaling factor $\gamma$ is an adjustable hyper-parameter.

*3.2.4* **Model Joint Training of MMKD.** We train our recommender using a multi-task learning scheme to jointly optimize MMKD with the following tasks: i) the main user-item interaction prediction task, represented by $\mathcal{L}_{\text{BPR}}$; ii) the pair-wise robust ranking KD $\mathcal{L}_{PairKD}$; iii) the modality-aware list-wise disentangled KD $\mathcal{L}_{ListKD}$; iv) modality-aware embedding KD $\mathcal{L}_{EmbKD}$. The overall

loss function $\mathcal{L}$ is given as follows:

$$\mathcal{L} = \mathcal{L}_{\text{BPR}} + \lambda_2 \cdot \mathcal{L}_{\text{PairKD}} + \lambda_3 \cdot \mathcal{L}_{\text{ListKD}} + \lambda_4 \cdot \mathcal{L}_{\text{EmbKD}} \quad (12)$$

$$\mathcal{L}_{\text{BPR}} = \sum_{u,i^+,i^-}^{|\mathcal{E}_{\text{bpr}}|} -\log\left(\text{sigmoid}(\mathbf{e}_u \cdot \mathbf{e}_{i^+} - \mathbf{e}_u \cdot \mathbf{e}_{i^-})\right) + \|\Theta\|^2 \quad (13)$$

where $\lambda_2$, $\lambda_3$, and $\lambda_4$ are parameters for loss term weighting. The last term $\|\Theta\|^2$ is the weight-decay regularization against over-fitting.

## 3.3 Model Complexity Analysis

The time complexity of current state-of-the-art graph-based multimodal recommender mainly consists of two parts: i) Modality feature reduction layer $\mathcal{R}(\cdot)$: The multimodal recommendation models inevitably need to incorporate feature reduction layers, as shown in Tab. 1. Most models employ a linear layer $O(\sum_{m \in \mathcal{M}} \times |\mathcal{I}| \times d_m \times d)$ or MLP transformation $O(\sum_{m \in \mathcal{M}} \times |\mathcal{I}| \times (d_m + d) \times d_h)$. $|\mathcal{I}|$ is the number of items. However, our inference model avoids the densification layer, due to the developed multi-modal knowledge distillation recommendation framework. ii) GNNs operations: Our inference model utilizes the LightGCN architecture solely in the graph convolutional component, resulting in the lowest consumption level $O(L \times |\mathcal{E}| \times d)$ among current graph-based recommendation models, where $L$ is the number of GNNs layers and $|\mathcal{E}|$ denotes the number of observed interactions. Other models (*e.g.*, LATTICE, SLMRec, MICRO) also use lightweight architectures. However, GRCN and LATTICE require reconstruction operations that consume $|\mathcal{I}|^2 \times d$ and $|\mathcal{I}|^2 \times d_m$, respectively. The difference between them is that the weights of the reconstructed edges are based on densification $d$ and the original high dimension $d_m$, respectively. LATTICE also takes $O(k \times |\mathcal{I}| \times log(|\mathcal{I}|))$ to retrieve top-$k$ most similar items for each item. We summarize the computational complexity of the graph-based multimodal methods in Tab. 1

## 4 EVALUATION

### 4.1 Experimental Settings

*4.1.1* **Dataset.** We conduct experiments on three multi-model recommendation datasets and summarize their statistics in Tab. 3.

- **Netflix:** This dataset contains user-item interactions from the Netflix platform. To construct the multi-model content, we crawled the movie posters based on the provided movie titles. The CLIP-ViT model [47] was used as the image feature extractor and BERT [27] is pre-trained for text feature encoding. We have released our pre-processed Netflix dataset, which includes the posters, to facilitate further research.

- **Tiktok:** This micro-video dataset [60] contains interactions with three types of modality features: visual, acoustic, and textual. The 128-dimensional visual and acoustic features were extracted from micro-videos desensitization, while the textual features were extracted from the captions using the Sentence-BERT model [48].

- **Electronics:** This dataset is based on the Electronics review data from Amazon. The visual modality includes 4,096-dimensional features that were extracted using pre-trained convolutional neural networks [15]. For the textual modality, we utilized Sentence-BERT [48] to combine various item attributes, such as title, descriptions, categories, and brands, into a compact 1024-d vector.

**Table 3: Statistics of experimented datasets with multi-modal item Visual (V), Acoustic (A), Textual (T) contents.**

| Dataset | Netflix | | Tiktok | | | Electronics | |
|---|---|---|---|---|---|---|---|
| Modality | V | T | V | A | T | V | T |
| Feat. Dim. | 512 | 768 | 128 | 128 | 768 | 4096 | 1024 |
| User | 43,739 | | 14,343 | | | 41,691 | |
| Item | 17,239 | | 8,690 | | | 21,479 | |
| Interaction | 609,341 | | 276,637 | | | 359,165 | |
| Sparsity | 99.919% | | 99.778% | | | 99.960% | |

* Tiktok: https://www.biendata.xyz/competition/icmechallenge2019/
* Electronics: http://jmcauley.ucsd.edu/data/amazon/links.html

*4.1.2* **Evaluation Protocols.** We use two widely adopted metrics for top-K item recommendation task: Recall@K (R@K) and Normalized Discounted Cumulative Gain (N@K). We set K to 20 and 50 to evaluate the performance of our approach and several state-of-the-art baselines. We adopted the all-ranking strategy for evaluation, following the settings used in previous works [61, 63]. To conduct significance analysis, $p$-values were calculated using the results of our proposed approach and the best-performing baseline.

*4.1.3* **Hyperparameter Settings.** We implemented our model framework using PyTorch and initialized model parameters using the Xavier initializer. We employ the AdamW optimizer [42] for both teacher $\mathcal{T}(\cdot)$ and student $\mathcal{S}(\cdot)$. The optimizer of the student will simultaneously optimize the parameters of both the student $\mathcal{S}(\cdot)$ and the prompt module $\mathcal{P}(\cdot)$, which is similar to that of the teacher's optimization. We search for the learning rates of $\mathcal{T}(\cdot)$ and $\mathcal{S}(\cdot)$ within the ranges of $[3.5e^{-4}, 9.8e^{-3}]$ and $[2.5e^{-4}, 8.5e^{-4}]$ respectively. The decay of the $L_2$ regularization term is tuned from $\{2.5e^{-3}, 7.4e^{-3}, 2.1e^{-2}\}$ for three datasets. All baselines are evaluated based on their source code and original papers, and the corresponding parameter tuning is conducted under a unified process.

*4.1.4* **Baselines.** To comprehensively evaluate the performance of our proposed approach, we compared it against several state-of-the-art baselines from different research lines.

i) **Collaborative Filtering Models**

- **BPR-MF** [49]: It presents a generic optimization criterion, BPR-Opt, for personalized ranking that outperforms standard learning techniques for matrix factorization and adaptive kNN.

- **NGCF** [58]: It introduces GNNs to CF framework to model high-order information. The newly proposed embedding propagation layer allows the embeddings of users and items to interact with long-range information to harvest the collaborative signal.

- **LightGCN** [17]: It simplifies the graph convolution to remove the transformation and activation modules for model simplification.

ii) **Multi-Modal Recommender Systems**

- **VBPR** [16]: It proposes a Matrix Factorization approach to incorporate visual signals into prediction of user's preference for personalized ranking with implicit feedback.

- **MMGCN** [64]: It is built upon the graph-based information propagation framework with a multi-modal GNN, so as to guide representation learning of user preference in each modality.

- **GRCN** [63]: It designs adaptive refinement module to identify and prune potential false positive edges in the interaction structure, by considering multi-modal item characteristics.

- **LATTICE** [74]: This method discovers latent relationships between modalities using modality-aware structure learning layers to supplement collaborative signals for recommendation.
- **CLCRec** [62]: It studies the cold-start recommendation task and maximizes the mutual dependencies between item content and collaborative signals using contrastive learning.
- **SLMRec** [53]: This work captures multi-modal patterns in data by generating multiple views of individual items and using contrastive learning to distill additional supervised signals.
- **BM3** [80]: It leverages a latent representation dropout mechanism instead of graph augmentation to generate the target view of users/items for contrastive task without negative samples.

## 4.2 Performance Comparison

Tab. 4 presents the results of all methods on three datasets, with the results of MMKD and the best baseline highlighted in bold and underlined, respectively. Based on results and our analysis, we make the following key observations and conclusions:

- The proposed MMKD consistently outperforms both general collaborative filtering (CF) models and state-of-the-art multi-modal recommendation methods on all three datasets, demonstrating its effectiveness in multimedia recommendation. In particular, our proposed approach outperforms the best baselines by 9.75%, 13.26%, and 11.79% in terms of NDCG@50. The improved outcomes are attributed to our designed multi-modal knowledge distillation enhanced by prompt-tuning, which not only bridges the semantic gap during the multi-modal knowledge transfer, but also eliminates the impact of noise and redundancy of modality data. Furthermore, our results support the idea that multi-modal recommender systems perform better than general CF models, due to the incorporation of multi-modal context for assisting user preference learning under sparse data.
- Our MMKD achieves competitive results with a lightweight architecture and tailored transferred knowledge, suggesting that there may be noise in the multi-modal data. This finding confirms our motivation that directly incorporating multi-modal information into the user representations may introduce noise, which can misguide the encoding of modality-aware user preferences. To address this issue, our proposed approach disentangles the soft labels of collaborative relations during the knowledge distillation, which effectively alleviates the noise of multi-modal content by transferring more informative signals into the student model.
- Multi-modal recommendation methods often exhibit significant performance fluctuations on different datasets, due to overfitting. These models are highly influenced by the quality of model features as well as the number of interactions. For instance, LATTICE performs worse on Netflix with many interactions, which we attribute to the introduction of noise by the homogeneous co-current graph. In contrast, GRCN achieves superior performance on Netflix by identifying and removing false-positive edges in user-item graphs. BM3 and CLCRec do not use classical negative sampling of BPR and perform better on datasets with more implicit feedback than on Tiktok and Electronics. We speculate that this is because negative samples do not necessarily indicate users' dislikes. It may simply be due to the item not being presented.

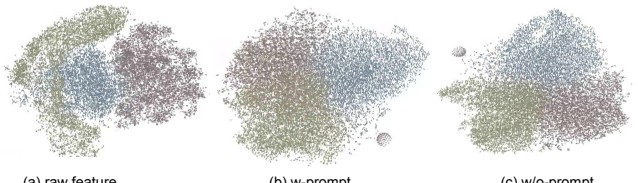

(a) raw feature      (b) w-prompt      (c) w/o-prompt

**Figure 3: t-SNE Visualization on Tiktok for raw high dimensional multi-modal features $\mathbf{X}^m$, modality-specific representations $\mathbf{F}^m$ of MMKD and $\mathbf{F}^m$ of variant $w/o$-Prompt.**

## 4.3 Ablation and Effectiveness Analyses

To justify the effectiveness of the proposed key components, we designed four variants of our MMKD and compared their performance against the original approach. The results in terms of Recall@20 and NDCG@20 are shown in Table 5. Further convergence analysis is provided in Supplementary. In order to gain deeper insights into the efficacy of the key components, we also conducted a further visualization analysis. Variant details are presented below:

- $w/o$-Prompt: This variant disables the prompt-tuning module to evaluate its impact on bridging the semantic gap during the teacher-student knowledge distillation process.
- $w/o$-PairKD: This variant examines the effect of ranking-based distillation for collaborative knowledge by removing the pairwise knowledge distillation loss term $\mathcal{L}_{\text{PairKD}}$ from the joint loss.
- $w/o$-ListKD: The modality-aware disentangled knowledge distillation is not included to re-weight the soft-labels for alignment with the fine-grained knowledge decoupling.
- $w/o$-disentangle: This variant preserves the list-wise distillation in Sec. 3.2.2, while removing the disentangled part. Aiming to validate the utility of extracting more informative signals from modality features $\mathbf{f}^m$ with the list-wise objective, as well as the necessity of decoupling the transferred knowledge.

*4.3.1* **Numerical Results.** As can be seen in the Tab. 5: (1) For variant $w/o$-Prompt, its performance on all three datasets has decreased compared to MMKD. This suggests that the removal of prompt-tuning may lead to the semantic gap for knowledge distillation. The modality-aware projection may also be in an overfitting state and can be limited to encode recommendation task-relevant multi-modal context without prompt-tuning enhancement. (2) The variant $w/o$-PairKD shows a decrease in performance compared to MMKD when pair-wise KD is disabled, demonstrating the strength of $\mathcal{L}_{\text{PKD}}$ in distilling ranking-based signals for model alignment. (3) Modality-aware list-wise distillation can finely extract quality modality-aware collaborative relationships, which helps in multi-modal recommendation. Therefore, the variant $w/o$-ListKD is inferior to the MMKD results. (4) The item-centric modality features are heavily biased against the preferences of the user. As a result, the variant $w/o$-disentangle performs poorly without disentangling and re-weighing distilled soft labels.

*4.3.2* **Visualization Analysis.** As shown in Fig. 3, We conducted a visual analysis of modality-specific features on the TikTok dataset to intuitively understand the influence of introducing prompt-tuning for bridging the teacher model and the student model. Specifically, we applied t-SNE with PCA initialization to reduce the dimensionality of both the modality-specific densified features $\mathbf{f}_i^m \in \mathbb{R}^{|I| \times d}$ ($w-$, $w/o-$ prompt-tuning) obtained from the feature densification layer, and the original multi-modal high-dimensional

**Table 4: Performance comparison of baselines on different datasets in terms of *Recall*@20/50, and *NDCG*@20/50.**

| Baseline | Netflix | | | | Tiktok | | | | Electronics | | | |
|---|---|---|---|---|---|---|---|---|---|---|---|---|
| | R@20 | N@20 | R@50 | N@50 | R@20 | N@20 | R@50 | N@50 | R@20 | N@20 | R@50 | N@50 |
| MF-BPR | 0.1583 | 0.0578 | 0.2396 | 0.0740 | 0.0488 | 0.0177 | 0.1038 | 0.0285 | 0.0211 | 0.0081 | 0.0399 | 0.0117 |
| NGCF | 0.1617 | 0.0612 | 0.2455 | 0.0767 | 0.0604 | 0.0206 | 0.1099 | 0.0296 | 0.0241 | 0.0095 | 0.0417 | 0.0128 |
| LightGCN | 0.1605 | 0.0609 | 0.2449 | 0.0768 | 0.0612 | 0.0211 | 0.1119 | 0.0301 | 0.0259 | 0.0101 | 0.0428 | 0.0132 |
| VBPR | 0.1661 | 0.0621 | 0.2402 | 0.0729 | 0.0525 | 0.0186 | 0.1061 | 0.0289 | 0.0234 | 0.0095 | 0.0409 | 0.0125 |
| MMGCN | 0.1685 | 0.0620 | 0.2486 | 0.0772 | 0.0629 | 0.0208 | 0.1221 | 0.0305 | 0.0273 | 0.0114 | 0.0445 | 0.0138 |
| GRCN | 0.1762 | 0.0661 | 0.2669 | 0.0868 | 0.0642 | 0.0211 | 0.1285 | 0.0311 | 0.0281 | 0.0117 | 0.0518 | 0.0158 |
| LATTICE | 0.1654 | 0.0623 | 0.2531 | 0.0770 | 0.0675 | 0.0232 | 0.1401 | 0.0362 | 0.0340 | 0.0135 | 0.0641 | 0.0184 |
| CLCRec | 0.1801 | 0.0719 | 0.2789 | 0.0892 | 0.0657 | 0.0214 | 0.1329 | 0.0329 | 0.0300 | 0.0118 | 0.0559 | 0.0169 |
| SLMRec | 0.1743 | 0.0682 | 0.2878 | 0.0869 | 0.0669 | 0.0221 | 0.1363 | 0.0342 | 0.0331 | 0.0132 | 0.0624 | 0.0180 |
| BM3 | 0.1792 | 0.0720 | 0.2842 | 0.0923 | 0.0660 | 0.0225 | 0.1351 | 0.0343 | 0.0336 | 0.0141 | 0.0637 | 0.0195 |
| MMKD | **0.1864** | **0.0743** | **0.3054** | **0.1013** | **0.0737** | **0.0258** | **0.1517** | **0.0410** | **0.0369** | **0.0155** | **0.0691** | **0.0218** |
| *p*-value | $1.60e^{-6}$ | $5.90e^{-5}$ | $2.99e^{-7}$ | $1.11e^{-6}$ | $1.41e^{-4}$ | $5.59e^{-4}$ | $5.00e^{-6}$ | $1.29e^{-5}$ | $3.24e^{-5}$ | $2.96e^{-6}$ | $7.51e^{-7}$ | $4.63e^{-6}$ |

**Table 5: Ablation study on key components of MMKD**

| Data | Netflix | | Tiktok | | Electronics | |
|---|---|---|---|---|---|---|
| Metrics | R@20 | N@20 | R@20 | N@20 | R@20 | N@20 |
| *w/o*-Prompt | 0.1665 | 0.0662 | 0.0681 | 0.0240 | 0.0280 | 0.0117 |
| *w/o*-PairKD | 0.1774 | 0.0689 | 0.0692 | 0.0242 | 0.0277 | 0.0112 |
| *w/o*-ListKD | 0.1690 | 0.0487 | 0.0673 | 0.0234 | 0.0331 | 0.0136 |
| *w/o*-disentangle | 0.1712 | 0.0693 | 0.0706 | 0.0249 | 0.0353 | 0.0141 |
| MMKD | **0.1864** | **0.0743** | **0.0737** | **0.0258** | **0.0369** | **0.0155** |

**Table 6: Model compactness and inference efficiency. "Time" indicates the average recommendation time for each epoch. "Memory" represents GPU memory usage. "Params" denotes the number of parameters. "Ratio" indicates the relative parameter size compared to the teacher. We use PyTorch with CUDA from RTX 3090 GPU and Intel Xeon W-2133 CPU.**

| Dataset | T-Model | Time | Memory | # Params | Ratio |
|---|---|---|---|---|---|
| Netflix | Teacher | 42.6s | 2.95GB | 24.91M | - |
| | LATTICE | 61.0s | 18.24GB | 24.06M | 96.59% |
| | BM3 | 39.5s | 2.69GB | 24.06M | 96.59% |
| | MMKD | 23.3s | 2.03GB | 1.95M | 7.83% |
| Electronics | Teacher | 30.8s | 5.02GB | 99.04M | - |
| | LATTICE | 45.1s | 37.69GB | 98.39M | 99.34% |
| | BM3 | 31.8s | 4.57GB | 98.39M | 99.34% |
| | MMKD | 13.9s | 3.81GB | 2.67M | 2.70% |

* LATTICE out of memory on Electronics dataset, and we completed its experiment on A100.

features $\mathbf{x}^m \in \mathbb{R}^{|I| \times d_m}$ into a 2-dimensional space. The results show that the original features $\mathbf{x}^m$ of diverse modalities exhibit significant differences in their vector space representation, with clear distinctions among different modalities, highlighting their association with distinct distributions. For modality-specific features $\mathbf{f}_i^m$, there are more overlaps in the prompt-tuning version, while the non-prompt-tuning version $w/o-$Prompt remains more confined to a modality-specific space. This suggests that prompt-tuning effectively strengthens the encoding of modality-specific features by extracting common user preferences pertaining to multiple ranking tasks while reducing the task-irrelevant features characteristic.

## 4.4 Study on Resource Consumption

In this section, we investigate the resource utilization of the teacher, student, and several baselines (BM3 and LATTICE) in terms of training time, storage, parameter count, and student-to-teacher

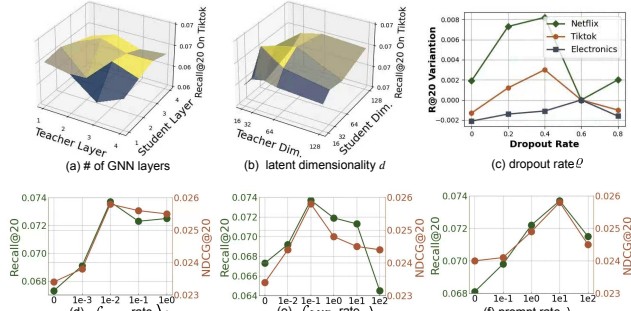

**Figure 4: Impact study of hyperparameters in MMKD.**

parameter ratio for model compression. The specific numerical results on Netflix and Electronics are reported in Tab. 6. Results show that Our student model exhibits significantly lower inference and recommendation time consumption than other models, likely due to their larger size, which requires more time during gradient descent parameter updates. Additionally, BM3 has a self-supervised joint training task, and LATTICE has to dynamically learn homogeneous graphs, which increase computational time consumption. We find that the calculation of KL-Divergence in our model does not significantly increase time consumption, resulting in lower latency.

Moreover, the results show that our model has low storage consumption, with a much lower parameter quantity compared to other models, such as LATTICE which needs to dynamically calculate and store item-time relationships, incurring significant overhead. The numerical value of 'ratio=11.24% or 2.70%' indicates the effectiveness of our model as a compression algorithm. Supplementary provides model evaluation results with online incremental learning.

## 4.5 Impact Study of Hyperparameters

This section investigates the influence of several important hyperparameters in our proposed MMKD. We report the evaluation results in Fig. 4 and examine the effect of one hyperparameter at a time while keeping other parameters at their default settings.

- Representation Dimensionality $d$: We investigated the influence of representation dimensionality $d$ on both the student $\mathcal{S}(\cdot)$ and teacher $\mathcal{T}(\cdot)$, with respect to the impact on recommendation system outcomes. We selected values of $d$ from [16, 32, 64, 128], and found that the model's performance saturates when the

number of hidden units reaches approximately 64 for the student. Notably, when the dimensions of the teacher and student are the same, the student's results are better. This is because the score of KD is obtained by the inner product of the representations, and the dimension size determines the scale of the score. Having the same scale level leads to more accurate KD.

- Depth of GNNs $L$: We examine the influence of the depth of the GNNs in the range of [1, 2, 3, 4]. The results show that the teacher's performance improves as the layer count increases, while the student's performance remains moderate. We speculate that this is because the teacher needs to encode useful knowledge with high-order relationships, and our modality-aware ranking-based KD effectively transfers quality knowledge to the student.

- Dropout Ratio of Teacher's Modality Encoding Layer: We investigate the influence of the dropout ratio of the teacher's modality encoding layer, which ranges from 0 to 1. Our results show that without dropout, the teacher's performance drops sharply, indicating overfitting in the multi-modal feature encoder. A higher dropout rate is required for datasets with higher original feature dimensions, confirming the risk of overfitting the multi-modal feature with existing modality encoders.

- Pair/List-wise KD Loss Weight $\lambda_2, \lambda_3$: The pair/list-wise KD loss weights ($\lambda_2, \lambda_3$) indicate the strength of collaborative knowledge distillation and disentangled modality-aware knowledge distillation, respectively. We vary the weights in the range of [0, 1e-2, 1e-1, 1e0, 1e1, 1e2]. Evaluation results show that absent or small weights significantly decrease the model's performance.

- Prompt Rate $\lambda_1$: It controls the soft-token rate. Our results show that without a soft-token rate, the model's performance significantly decreases, indicating that prompt-tuning enables teachers to generate more helpful knowledge for students. We speculate that this is because the prompt module optimizes alongside the student learning, leading to better recommendation performance.

## 5 RELATED WORK

**Multi-Modal Recommender Systems**. Researchers have explored using multi-modal content to enhance recommenders. Pioneering studies, such as VBPR[16] have incorporated visual features into CF frameworks. After that, GNN-enhanced multi-modal recommenders capture high-order connectivity by incorporating modality signals. Inspired by the success of self-supervised learning, recent multimedia recommenders, such as MMGCL [73], SLMRec [53], use data augmentation strategies to enhance the representation learning, PAMD[13] uses contrastive learning to align modalities. Additionally, some other works focus on a different scenario for recommending multimedia content (*e.g.*, fashion [66], news[50, 65], micro-video [23, 38]) with sequential learning [34, 79]. For the cold start scenario, several studies (*e.g.*, MTPR [4], MML [45], and INP [36]) have used multi-modal content to improve item representations. For debiasing in multi-modal scenario, CausalRec [46] uses a causal graph for visual debiasing while InvRL[5], CGKR[44] addresses spurious correlations in multimedia content. Despite their effectiveness, most of them are built upon cumbersome multi-modal feature encoding which limits their scalability in practice.

**Knowledge Distillation for Recommendation**. KD in recommendation has sparked various research directions [54, 68]. Many studies have explored the distillable aspects of recommenders. For relation-based KD, HTD[25] distilled the topological knowledge built upon the relations in the teacher embedding space and [78] distilled structured knowledge from a differentiable path-based recommendation model. DE-RRD[24] conducts both feature-based KD and relation-based KD. Besides, different distillation paradigms are also of interest to researchers[71, 72]. LUME[71] uses multi-teacher ensemble and debiased KD to aggregate knowledge from multiple pretrained teachers. CrossDistil[72], [30] encourage mutual distillation between models. KD can bring benefits for scenario-specific recommenders[26, 52, 57]. For the recommender with heterogeneous information, HetComp[26] transfers the ensemble knowledge of heterogeneous teachers to a lightweight student and DESIGN[52] integrates information from the homogenous graph and trains two auxiliary smaller models. Recommender with privileged feature also can leverage KD[57]. For example, student of PGD[57] can learn the distilled output with privileged CF embeddings to tackle the cold start problem. And there also are some efforts focus on debias when conducting KD[2, 71].

We introduce KD in multi-modal scenario for two purposes: i) Solve the problem of large coding parameter models caused by high-dimensional output features in multi-modal scenes; ii) reduce the impact of noise in modal content and emphasize the transfer of knowledge that is useful and relevant for downstream tasks.

**Prompt Learning**. Prompt learning has become a emerging research direction in the context of large pre-trained models [1, 37]. It allows these models to be repurposed for different tasks without requiring additional training [39, 43]. The core idea behind prompt learning is to generate effective prompts that guide the model to generate the expected output for specific downstream tasks [12, 31]. Recent studies explores prompt learning in various domains, such as graph mining, knowledge graph representation learning and recommendation [8, 40, 76]. For example, GraphPrompt [41] defines the paradigm of prompts on graphs. To transfer knowledge graph semantics into task data, KGTransformer [77] regards task data as a triple prompt for tuning. Additionally, prompt-based learning has also been introduced to benefit recommender systems to enhance model explanation [32], fairness [67], sequence learning [69]. Motivated by these research lines, we propose a novel multi-modal prompt learning approach that can adaptively guide knowledge distillation for simple yet effective multimedia recommendation.

## 6 CONCLUSION

The objective of this work is to simplify and enhance multi-modal recommenders using a novel modality-aware KD framework empowered by prompt-tuning. In order to effectively transfer task-relevant knowledge from the teacher to the student model, we introduce a learnable prompt module that dynamically bridges the semantic gap between the multi-modal context encoding in the teacher model and the collaborative relation modeling in the student model. Additionally, our proposed framework, called MMKD, aims to disentangle the informative collaborative relationships, thereby enabling augmented knowledge distillation. Through extensive experiments, we demonstrate that MMKD significantly improves model efficiency while maintaining superior accuracy compared to state-of-the-art solutions. Our future work plans to integrate LLMs with multi-modal context encoding for performance enhancement.

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

# 7 SUPPLEMENTARY MATERIAL

In our supplementary materials, we begin by outlining the key steps involved in learning our proposed model. We then provide detailed derivations of the decoupling collaborative relationships to supplement our model's ability to distill more informative signals and reduce noise in the multi-modal knowledge distillation paradigm. Furthermore, we present additional experiments on real-time incremental learning scenarios to further demonstrate the validity of our proposed MMKD framework in practical scenarios. Finally, we provide a list of the source code used for the baseline methods.

## 7.1 Pseudocode of MMKD's Learning Procedure

In this subsection, we outline the key steps involved in learning our proposed MMKD with the following presented pseudocode:

---

**Algorithm 1:** Framework of MMKD

---

**Input:** Bipartite interaction graph G=({U, I},E, **X**), with modality-specific node attribute $\mathbf{X}^m \in \mathbb{R}^{|I| \times d_m}(\mathbf{X}^m \in \mathbf{X})$ and normalized user-item interaction matrix **A**.

**Output:** User's modality-aware interactive preference $\hat{y}_{u,i}(i \in I, u \in \mathcal{U})$.

1 **Initialize:** Xavier initialized parameters of $\mathcal{T}(\cdot), \mathcal{S}(\cdot), \mathcal{P}(\cdot), \mathcal{R}(\cdot)$.
   // Offline Teacher Training
2 **for** $epoch \leftarrow 0, 1, \ldots$ **do**
3    **for** $step \leftarrow 0, 1, \ldots$ **do**
      // Teacher Inference
4       $\mathbf{p} \longleftarrow$ Eq. 3
5       $\mathbf{E}_u^{\mathcal{T}}, \mathbf{E}_i^{\mathcal{T}}, \quad \mathbf{F}_u^1, ..., \mathbf{F}_u^m, ..., \mathbf{F}_i^1, ..., \mathbf{F}_i^m... \longleftarrow \mathcal{T}(\mathbf{A}, \mathbf{X})$
      // Teacher Objective
6       $\mathcal{L}_{\text{BPR}} \longleftarrow \mathbf{E}_u^{\mathcal{T}}, \mathbf{E}_i^{\mathcal{T}}$
7       Gradient descent for $\mathcal{T}(\cdot)$ and $\mathcal{P}(\cdot)$.
8    **end**
9 **end**
   // Student Training
10 **for** $epoch \leftarrow 0, 1, \ldots$ **do**
11    **for** $step \leftarrow 0, 1, \ldots$ **do**
      // Teacher Inference (as above)
12       ...
      // Student Inference
13       $\mathbf{E}_u^{\mathcal{S}}, \mathbf{E}_i^{\mathcal{S}} \longleftarrow \mathcal{S}(\mathbf{E}_u^{\mathcal{T}}, \mathbf{E}_i^{\mathcal{T}}, \mathbf{A})$
      // Knowledge Distillation
14       $\mathcal{L}_{\text{PairKD}} \longleftarrow \mathbf{E}_u^{\mathcal{T}}, \mathbf{E}_i^{\mathcal{T}}, \mathbf{E}_u^{\mathcal{S}}, \mathbf{E}_i^{\mathcal{S}}$
15       $\mathcal{L}_{\text{ListKD}} \longleftarrow \mathbf{F}_u^1, ..., \mathbf{F}_u^m, ..., \mathbf{F}_i^1, ..., \mathbf{F}_i^m..., \mathbf{E}_u^{\mathcal{S}}, \mathbf{E}_i^{\mathcal{S}}$
16       $\mathcal{L}_{\text{EKD}} \longleftarrow \mathbf{F}_u^1, ..., \mathbf{F}_u^m, ..., \mathbf{F}_i^1, ..., \mathbf{F}_i^m..., \mathbf{E}_u^{\mathcal{S}}, \mathbf{E}_i^{\mathcal{S}}$
      // Student Objective
17       L
      $\longleftarrow \mathcal{L}_{\text{BPR}} + \lambda_2 \cdot \mathcal{L}_{\text{PairKD}} + \lambda_3 \cdot \mathcal{L}_{\text{ListKD}} + \lambda_4 \cdot \mathcal{L}_{\text{EmbKD}}$
18       Gradient descent for $\mathcal{S}(\cdot)$ and $\mathcal{P}(\cdot)$.
19    **end**
20 **end**

---

## 7.2 Derivations of Disentangling Ranking Score

In our MMKD, we propose to improve our collaborative knowledge distillation approach by incorporating modality-aware user interaction patterns and decoupling the transferred knowledge based on its degree of uncertainty. By considering the reliability of observed user-item relations, including interactions and non-interactions, we can better identify the most informative collaborative signals from the teacher model. To achieve this, we will utilize a modality-aware decoupled KD with a listwise loss, which will allow us to more flexibly handle collaborative signals related to different modalities.

The details presented below are related to the mathematical derivation discussed in Section 3.2.2:

$$KL(\mathbf{y}_{\text{list}}^T \| \mathbf{y}_{\text{list}}^S) = \sum_{k=1}^{K} y_k^T log(\frac{y_k^T}{y_k^S}) \tag{14}$$

$$= y_1^{+T} log(\frac{y_1^{+T}}{y_1^{+S}}) + \sum_{k=2,k\neq 1}^{K} y_k^{-T} log(\frac{y_k^{-T}}{y_k^{-S}})$$

$$= b^{+T} log(\frac{b^{+T}}{b^{+S}}) + \sum_{k=2,k\neq 1}^{K} y_k^{-T} log(\frac{y_k^{-T}}{y_k^{-S}})$$

Eq.12 can be expressed as $q_k = y_k/b^-$ by applying Eq. 6 and Eq. 7, as the class index $k$ is not affected by $b^{-T}$ and $b^{-S}$.

$$= b^{+T} log(\frac{b^{+T}}{b^{+S}}) + \sum_{k=2,k\neq 1}^{K} y_k^{-T} log(\frac{y_k^{-T}}{y_k^{-S}}) \tag{15}$$

$$= b^{+T} \log(\frac{b^{+T}}{b^{+S}}) + \sum_{k=2,k\neq 1}^{K} b^{-T} q_k^T (\log(\frac{b^{-T} q_k^T}{b^{-T} q_k^S}))$$

$$= b^{+T} \log(\frac{b^{+T}}{b^{+S}}) + \sum_{k=2,k\neq 1}^{K} b^{-T} q_k^T (\log(\frac{b^{-T}}{b^{-S}}) + \log(\frac{q_k^T}{q_k^S}))$$

$$= b^{+T} \log(\frac{b^{+T}}{b^{+S}}) + \sum_{k=2,k\neq 1}^{K} b^{-T} q_k^T \log(\frac{b^{-T}}{b^{-S}}) + \sum_{k=2,k\neq 1}^{K} b^{-T} q_k^T (\log(\frac{q_k^T}{q_k^S}))$$

$$= b^{+T} \log(\frac{b^{+T}}{b^{+S}}) + b^{-T} \log(\frac{b^{-T}}{b^{-S}}) \sum_{k=2,k\neq 1}^{K} q_k^T + \sum_{k=2,k\neq 1}^{K} b^{-T} q_k^T (\log(\frac{q_k^T}{q_k^S}))$$

$$= b^{+T} \log(\frac{b^{+T}}{b^{+S}}) + b^{-T} \log(\frac{b^{-T}}{b^{-S}}) + \sum_{k=2,k\neq 1}^{K} b^{-T} q_k^T (\log(\frac{q_k^T}{q_k^S}))$$

$$= b^{+T} \log(\frac{b^{+T}}{b^{+S}}) + b^{-T} \log(\frac{b^{-T}}{b^{-S}}) + (1 - b^{+T}) \sum_{k=2,k\neq 1}^{K} q_k^T (\log(\frac{q_k^T}{q_k^S}))$$

Consequently, $KL(\mathbf{y}_{\text{list}}^T \| \mathbf{y}_{\text{list}}^S)$ may be rewritten as follows:

$$= \underbrace{b^{+T} \log(\frac{b^{+T}}{b^{+S}}) + b^{-T} \log(\frac{b^{-T}}{b^{-S}})}_{KL(\mathbf{b}^T \| \mathbf{b}^S)} + \underbrace{(1 - b^{+T}) \sum_{k=2,i\neq 1}^{K} q_{k_k}^T \log(\frac{q_k^T}{q_k^S})}_{KL(\mathbf{q}^T \| \mathbf{q}^S)} \tag{16}$$

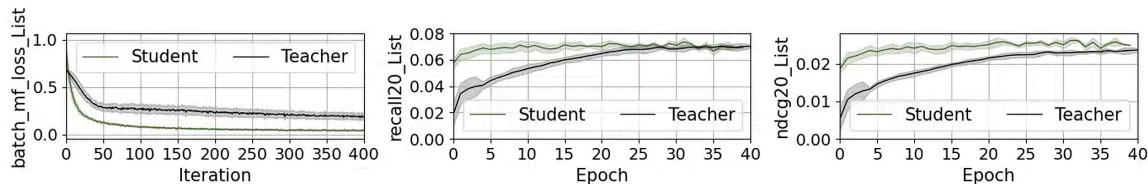

**Figure 5: Training curves of MMKD framework in terms of Recall@20, NDCG@20, and $\mathcal{L}$ on Tiktok dataset.**

By discriminating the knowledge transferred from the teacher model in terms of its uncertainty, our model can effectively manage the collaborative signals associated with different modalities, thereby reducing the noise and improving the accuracy of the transferred knowledge in our multi-modal knowledge distillation.

## 7.3 Supplementary Experiments

*7.3.1* **Incremental Learning over New Data.** To validate the efficiency and effectiveness of our lightweight inference model (*i.e.*, student) for making recommendation in real-life online platforms, we simulate a scenario where the model is fine-tuned with a small number of new interactions and used for predicting dynamic user preferences movies from Netflix data.

**Table 7: Performance of model adaptation to new data. The time cost (measured by ms) is calculated by training the model and making inference with recommendation results. Better results can be obtained by our model compared with baselines in terms of both efficiency and accuracy.**

| Methods | Recall@20 | NDCG@20 | Time |
|---------|-----------|---------|------|
| LATTICE | 0.1210 | 0.0487 | 1,605.26ms |
| BM3 | 0.1381 | 0.0574 | 790.0ms |
| MMKD | 0.1454 | 0.0586 | 482.5ms |

Table 7 demonstrates that the lightweight inference model outperforms the baseline methods (BM3 and LATTICE) with cumbersome multi-modal encoding frameworks in terms of efficiency. Our designed multi-modal knowledge distillation paradigm allows our model to achieve comparable recommendation accuracy to more complex multi-modal recommender systems (such as BM3), while significantly improving performance compared to LATTICE. This is because our model effectively transfers multi-modal collaborative relationships from the teacher model, thereby preserving modality-aware user preferences in the smaller student model.

The experiments performed under incremental learning recommendation environment indicate that a lightweight inference model can better adapt to new recommendation data with much lower computational cost, while performing comparably to more cumbersome baselines in terms of Recall@20 and NDCG@20. This further confirms the performance superiority of our MMKD in practical recommender systems, providing an efficient and effective inference model for real-time recommendation.

*7.3.2* **Convergence Analysis.** In this section, we use convergence analysis to examine the effects of our multi-modal knowledge distillation recommendation system on the effectiveness of model training. The convergence process with respect to Recall@20 and $\mathcal{L}$ for each epoch is shown in Fig. 5.

*7.3.3* **Parameter Size Comparison.** By analyzing the parameter statistics of both the teacher and student models, we can observe that our MMKDeffectively compresses the student model, resulting in a significantly smaller parameter size.

**Table 8: # Parameters of Teacher and Student Models.**

| # Parameters: Teacher (Netflix) | | |
|---|---|---|
| Weight Name | Weight Shape | Number |
| image_trans.weight | torch.Size([32, 512]) | 16384 |
| image_trans.bias | torch.Size([32]) | 32 |
| text_trans.weight | torch.Size([32, 768]) | 24576 |
| text_trans.bias | torch.Size([32]) | 32 |
| user_id_embedding.weight | torch.Size([43739, 32]) | 1399648 |
| item_id_embedding.weight | torch.Size([43739, 32]) | 1399648 |
| image_embedding.weight | torch.Size([17239, 32]) | 8826368 |
| text_embedding.weight | torch.Size([17239, 32]) | 13239552 |
| batch_norm.weight | torch.Size([32]) | 32 |
| batch_norm.bias | torch.Size([32]) | 32 |
| The total number of parameters: | | 24906304 |
| The parameters of Model Teacher_Model: | | 24.906304M |

| # Parameters: Student: LightGCN (Netflix) | | |
|---|---|---|
| Weight Name | Weight Shape | Number |
| user_id_embedding.weight | torch.Size([43739, 32]) | 1399648 |
| item_id_embedding.weight | torch.Size([17239, 32]) | 551648 |
| The total number of parameters: | | 1951296 |
| The parameters of Model Student_LightGCN: | | 1.951296M |

\* Correct the value in Tab.6.

| # Parameters: Student: GCN (Netflix) | | |
|---|---|---|
| Weight Name | Weight Shape | Number |
| layer_list.0.user_weight | torch.Size([32, 32]) | 1024 |
| layer_list.0.item_weight | torch.Size([32, 32]) | 1024 |
| The total number of parameters: | | 2048 |
| The parameters of Model Student_GCN: | | 0.002048M |

*7.3.4* **Baseline Model Implementations.** The publicly available source codes for the baselines can be found at the following URLs:

- **BPR-MF**: https://github.com/gamboviol/bpr.git
- **NGCF**: https://github.com/huangtinglin/NGCF-PyTorch.git
- **LightGCN**: https://github.com/kuandeng/LightGCN.git
- **VBPR**: https://github.com/DevilEEE/VBPR.git
- **MMGCN**: https://github.com/weiyinwei/MMGCN.git
- **GRCN**: https://github.com/weiyinwei/GRCN.git
- **LATTICE**: https://github.com/CRIPAC-DIG/LATTICE.git
- **CLCRec**: https://github.com/weiyinwei/CLCRec.git
- **SLMRec**: https://github.com/zltao/SLMRec.git
- **BM3**: https://github.com/enoche/BM3.git