# OpenReview forum: "Multi-Modal Knowledge Distillation for Recommendation with Prompt-Tuning"
_ACM.org/TheWebConf/2024/Conference — TheWebConf24_

### Official Review · Reviewer_Q5Hv · 2023-11-05

**Novelty:** 5
**Technical Quality:** 5

**Review:**

This manuscript discusses the challenges faced by multi-modal recommenders and proposes a solution called Multi-Modal Knowledge Distillation (MMKD) with prompt-tuning. The main challenges addressed are overfitting caused by high-dimensional multi-modal features and inaccuracies in media content that affect user preference modeling. MMKD aims to simplify and empower recommenders by conducting model compression through distilling edge relationship and multi-modal node content. It also introduces soft prompt-tuning to bridge the semantic gap between multi-modal context and collaborative signals. Additionally, a disentangled multi-modal list-wise distillation is developed to adjust the impact of inaccuracies in multimedia data. Experimental results demonstrate the superiority of MMKD over existing techniques. The main contributions of the work are the development of the MMKD framework, integration of prompt-tuning with multi-modal knowledge distillation, and evaluation of the model's performance on real-world datasets.

Strengths:
1. Multi-modal recommenders face challenges of overfitting and inaccuracies in media content.
2. MMKD proposes model compression through distilling edge relationship and multi-modal node content.
3. Soft prompt-tuning is introduced to bridge the semantic gap between multi-modal context and collaborative signals.
4. A disentangled multi-modal list-wise distillation is developed to adjust the impact of inaccuracies in multimedia data.
5. Experimental results demonstrate the superiority of MMKD over existing techniques.

Weakness:
1. The division information of the benchmark datasets for the experiments is unclear.

**Questions:**

1. Could you provide the division information of the benchmark datasets for the experiments?

**Reviewer Confidence:**

3: The reviewer is confident but not certain that the evaluation is correct

**Scope:**

4: The work is relevant to the Web and to the track, and is of broad interest to the community

---

### Official Review · Reviewer_x1qM · 2023-11-13

**Novelty:** 5
**Technical Quality:** 5

**Review:**

Pros:
Innovative Approach: The MMKD framework represents an innovative approach in the field of recommender systems, particularly in handling multi-modal data.
Model Compression: MMKD effectively compresses complex multi-modal models into more efficient forms, reducing resource consumption without sacrificing accuracy.

Cons:
Computational Resources: Despite its model compression, the initial training and setup of MMKD might still require significant computational resources, especially when dealing with large datasets.

How does the MMKD model mitigate the risk of overfitting when dealing with high-dimensional multi-modal features?

**Questions:**

In what ways does the MMKD model distinguish between relevant and irrelevant multimedia content for user preference learning?

What techniques are employed in the MMKD model to bridge the semantic gap between multi-modal context and user-item collaborative relations?

**Reviewer Confidence:**

3: The reviewer is confident but not certain that the evaluation is correct

**Scope:**

3: The work is somewhat relevant to the Web and to the track, and is of narrow interest to a sub-community

---

### Official Review · Reviewer_wkXN · 2023-11-22

**Novelty:** 4
**Technical Quality:** 5

**Review:**

Pros:

- The paper focuses on two problems in multimodal recommendation: large number of parameters with multimodal encoders and inaccuracies in multimodal information. Specifically, the authors propose conducts model compression through distilling u-i edge relationship and multi-modal node content, and introduce soft prompt-tuning to bridge the semantic gap between multimodal context and collaborative signals. The core contribution is novel and the methods are effective.
- The paepr is well written and the illustrations are clear and easy to understand.
- The experiment is very abundant.


Cons:
- The methods are a little complex.
- Since many models could be used as teacher model, it's better to conduct multiple experiments w.r.t. different teacher models.
- In the motivation, the authors claim that most models suffer from overfitting problem but use them as teacher models. How could we obtain a proper student model with using an overfitted teacher model?

**Questions:**

- The methods are a little complex.
- Since many models could be used as teacher model, it's better to conduct multiple experiments w.r.t. different teacher models.
- In the motivation, the authors claim that most models suffer from overfitting problem but use them as teacher models. How could we obtain a proper student model with using an overfitted teacher model?

**Reviewer Confidence:**

4: The reviewer is certain that the evaluation is correct and very familiar with the relevant literature

**Scope:**

4: The work is relevant to the Web and to the track, and is of broad interest to the community

---

### Official Review · Reviewer_6ciz · 2023-11-22

**Novelty:** 5
**Technical Quality:** 5

**Review:**

In this paper, the authors introduce the multimedia knowledge distillation with the prompt-tuning to address two critical challenges in multi-modal recommenders: the issue of overfitting caused by excessive parameters and the problem of preference deviation due to redundant factors. The main content of the paper is written very clearly, with well-crafted writing, comprehensive logic, and thorough experimentation.

Strengths:

1.	This paper proposes a multi-modal KD framework MMKD for multimedia recommendation, which is able to produce a lightweight yet effective student inference recommender with minimal online inference time and resource consumption. The approach appears technically sound.

2.	The experiments evaluating the proposed MMKD method against 10 baseline approaches on three benchmark datasets under various settings. This provides convincing empirical evidence for the efficacy of MMKD.

3.	This paper is well-written, with a rigorous organizational structure and clear figures. The introduction provides necessary background, the proposed method is explained in adequate detail, and the experimental setup and results are presented clearly.

Weaknesses:

1.	Some spelling and grammar issues need more attention.

**Questions:**

1.	The authors introduce soft prompt-tuning to perform student task-adaptive distillation for tackling redundancy. Compared to existing mainstream prompt-tuning methods, what sets MMKD apart and where does its advantage lie?

2.	The configuration of the SCE loss seems quite interesting. Have the authors conducted experiments to compare the results of this loss with MSE loss? Additionally, it appears that there is a lack of ablation experiments for embedding KD.

3.	Spelling and grammar issues need attention. For example, in section 3.1.2, "a sutdent-task related signals as a hint" should be corrected to "student-task." Additionally, in section 3.2.1, "are is taken as a logit of KD loss" requires clarification.

**Reviewer Confidence:**

4: The reviewer is certain that the evaluation is correct and very familiar with the relevant literature

**Scope:**

4: The work is relevant to the Web and to the track, and is of broad interest to the community

---

### Official Review · Reviewer_uwkR · 2023-11-30

**Novelty:** 5
**Technical Quality:** 5

**Review:**

The paper proposed a knowledge distillation framework for multimodal recommendation. The framework mainly consists of the joint multimodal contents and cf signals to perform the knowledge distillation. The experiment results show that the proposed framework outperforms the compared scheme in terms of different mertric.

**Questions:**

1. For I1 and I2 in the introduction section, is there any quantitative study for them, such as the overfitting curves or the feature visualizations for showing the semantic gaps?
2. In the method section, P is used to denote the prompt in 3.1.2 and the logit list in 3.2.2. Are they the same thing? It reads a bit confusing.

**Reviewer Confidence:**

2: The reviewer is willing to defend the evaluation, but it is likely that the reviewer did not understand parts of the paper

**Scope:**

2: The connection to the Web is incidental, e.g., use of Web data or API

---

### Decision · Program_Chairs · 2024-01-22

**Decision:**

Accept

**Comment:**

Based on the reviews and discussions, this paper has received generally positive comments for its novelty and technical quality. Reviewers agree the proposed MMKD framework can address some challenges in multi-modal recommendation, such as overfitting and inaccuracies in media content. Overall, the paper is seen as a valuable contribution to the field, with robust experimental validation and clear, well-structured writing. There still are concerns about the complexity of the methods and potential issues with using overfitted teacher models in the distillation process. Some reviewers also suggest additional experiments and clarification on certain aspects.